# Misalignment with the external light environment drives metabolic and cardiac dysfunction

Alexander C. West[1], Laura Smith[1], David W. Ray [1], Andrew S.I. Loudon[1], Timothy M. Brown[1] & David A. Bechtold [1]

Most organisms use internal biological clocks to match behavioural and physiological processes to specific phases of the day–night cycle. Central to this is the synchronisation of internal processes across multiple organ systems. Environmental desynchrony (e.g. shift work) profoundly impacts human health, increasing cardiovascular disease and diabetes risk, yet the underlying mechanisms remain unclear. Here, we characterise the impact of desynchrony between the internal clock and the external light–dark (LD) cycle on mammalian physiology. We reveal that even under stable LD environments, phase misalignment has a profound effect, with decreased metabolic efficiency and disrupted cardiac function including prolonged QT interval duration. Importantly, physiological dysfunction is not driven by disrupted core clock function, nor by an internal desynchrony between organs, but rather the altered phase relationship between the internal clockwork and the external environment. We suggest phase misalignment as a major driver of pathologies associated with shift work, chronotype and social jetlag.

[1] Division of Diabetes, Endocrinology and Gastroenterology, School of Medicine, Faculty of Biology, Medicine and Health, University of Manchester, Manchester Academic Health Science Centre, Manchester M13 9PL, UK. Correspondence and requests for materials should be addressed to D.A.B. (email: david.bechtold@manchester.ac.uk)

The periodic succession of night and day influences nearly all forms of life on earth. As a result, organisms have evolved internal circadian clocks capable of keeping near precise 24 h time. By tracking time internally, organisms adapt their biology to match cyclical fluctuations in the environment (e.g. light, food availability, predation) and thus respond optimally. In mammals, the circadian system consists of a network of tissue clocks housed across the body and coordinated by a central pacemaker located within the suprachiasmatic nucleus (SCN) of the anterior hypothalamus. Within all of these sites, the molecular clock machinery drives rhythmic transcriptional and metabolic pathways in a tissue-specific manner, a process critical to proper tissue function[1–3]. Altered circadian function, be it through genetic variation, lifestyle factors (e.g. chronic shift work, sleep restriction, nocturnal light exposure) or experimental perturbation (e.g. forced desynchrony) are linked to a wide range of pathogenic states from metabolic disease to cancer[4–9]. A further, more prevalent circadian insult occurs when the phase of our internal timing (chronotype) does not match with patterns of societal-driven activity (commonly referred to as social jet-lag). Indeed, large population and targeted cohort studies are now linking chronotype with a variety of health, psychiatric and life history variables[10–13]. Given that modern life has disturbed the natural temporal structure of our environment, it is critical that we understand the mechanisms which link clock desynchrony to pathophysiological outcomes.

In mammals, circadian timekeeping is centred on the feedback coupling of the transcriptional activators CLOCK and BMAL1, and repressors PERIOD, CRYPTOCHROME and REVERB. Much of our understanding of clock function has been defined through genetic ablation of these and additional constituent factors[14, 15]. Animal studies demonstrate that genetic disruption of individual core clock genes can have a severe impact on heath, ranging from disturbances in metabolism and inflammatory response, to altered bone formation and neu-rodegeneration[14, 16, 17]. However, due to pleotropic and/or developmental activity of the targeted clock genes, it is rarely possible to isolate the influence of clock timing per se from that of the ablated factor[18, 19]. Moreover, in the context of human health, misalignment with the environment is overwhelmingly the principal source of circadian disruption, rather than the relatively minor contributions of genetic disruption in clock gene function.

Despite human circadian desynchrony being widely acknowledged as deleterious to health, the reasons for increased risk remain unclear, and the pathways to disease undefined. Here, we recapitulate in mice the circadian misalignment that occurs during shift work and in human subjects with extreme chronotype[20–22]. We show that long-term housing of mice under light–dark (LD) cycles that do not match a normal 24 h cycle leads to pronounced physiological disturbance, including altered metabolic efficiency and substrate utilisation, and a profound depression of cardiac function, including significant prolongation of PR and QT intervals. Our study

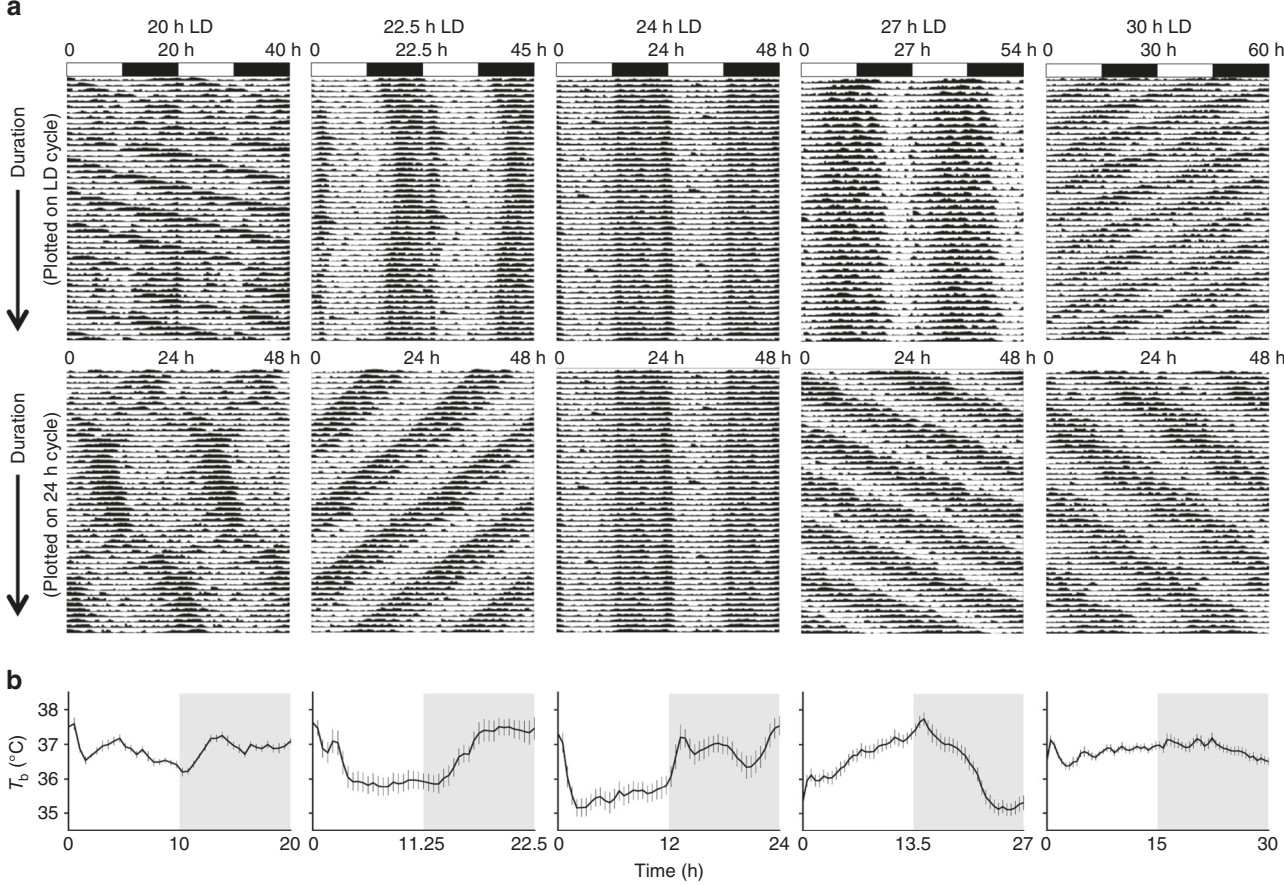

**Fig. 1** Long-term housing under non-24 h LD cycles drives phase desynchrony between physiological rhythms and the LD cycle. **a** Representative body temperature ($T_b$) rhythms from mice maintained on 20, 22.5, 24, 27 and 30 h LD cycles. *Top panels* are double plotted relative to LD cycle time; *bottom panels* are plotted on 24 h frequency. **b** Group $T_b$ profiles across the LD cycle (derived from 10 consecutive cycles, after 40 d LD exposure) highlight the delayed (22.5 h) and advanced (27 h) phase of entrainment in non-24 h LD housed mice. Data in **b** reflect mean ± SEM, $n = 7$/group for 20 and 30 h LD; $n = 10$/group for 22.5, 24 and 27 h LD

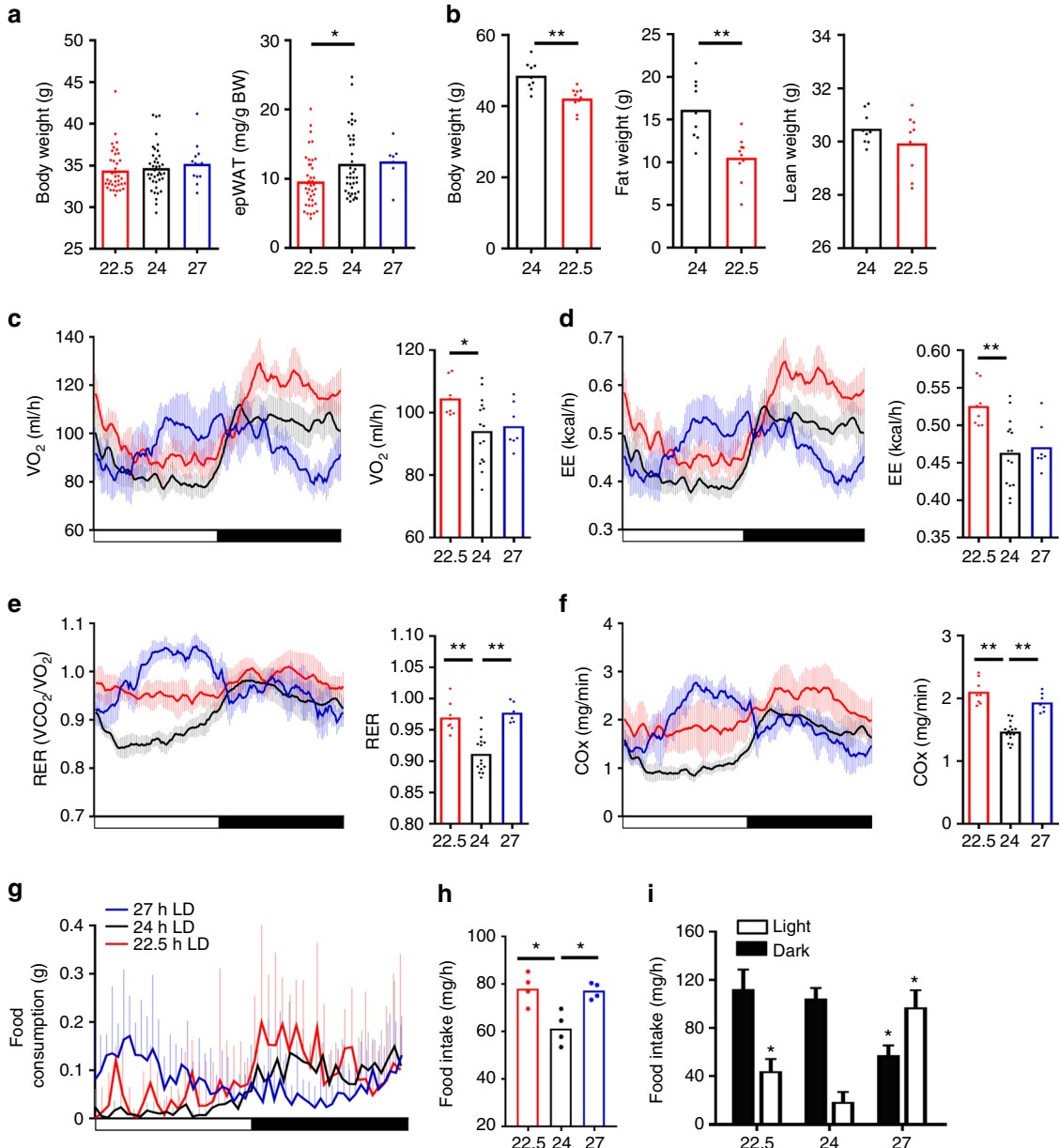

**Fig. 2** Phase misalignment drives reduced energy efficiency and metabolic disturbance. **a** Body weight and epididymal adipose tissue mass (epWAT) of mice maintained in 22.5, 24 or 27 h LD conditions for 17 week. epWAT was significantly lower in 22.5 h LD mice compared with age-matched 24 h controls ($n > 7$/group). **b** By 1 year of LD exposure, both body weight and whole-body fat mass were significantly lower in 22.5 h LD housed mice compared with matched 24 h LD mice ($n = 9$–10/group). **c–f** Diurnal rhythms and mean values for oxygen consumption (VO$_2$; **c**), energy expenditure (EE; **d**), respiratory exchange rate (RER; **e**) and carbohydrate oxidation (COx; **f**) for mice maintained under 22.5, 24 and 27 h LD cycles ($n = 7$–16/group). A significant increase in RER and COx was evident in non-24 h LD housed mice. **g–i** Diurnal profiles of food intake (**g**) and mean intake/h (**h**) reveal significantly increased food consumption in mice maintained under either 22.5 or 27 h LD cycles compared with 24 h LD control. Mice housed under non-24 h LD cycles also consumed more food during the light phase of the cycle (**i**). Diurnal profiles **c–g** reflect mean ± 95% CI. **a**, **c–h** *$p < 0.05$, **$p < 0.01$ one-way ANOVA with Dunnett's post hoc test (vs. 24 h control group); **b** **$p < 0.01$ Students $t$-test; **i** *$p < 0.05$ vs. 24 h LD two-way ANOVA with Tukey's post hoc test

highlights the importance of phase alignment of the circadian clock to the environment and implicates disrupted entrainment, common place in the modern world, as being a major driver of pathology.

## Results

**Entrainment to non-24 h light cycles leads to reduced energy efficiency.** To establish an altered phase relationship between the environmental LD cycle and the internal circadian clock, male C57Bl/6J mice were placed under 20 h, 22.5 h, 24 h, 27 h and 30 h symmetrical LD cycles for 17 week (from 8 week of age, $n > 8$/group). Mice maintained under 20 h or 30 h LD cycles were unable to entrain, and exhibited a free-running rhythm with a high degree of phase dispersion between individual animals (Fig. 1; Supplementary Fig. 1). In contrast, mice maintained in 22.5 h, 24 h and 27 h conditions achieved stable entrainment in locomotor activity and core body temperature ($T_b$). However, in comparison to 24 h LD housed mice, the phase of behavioural or physiological rhythms (as defined by the acrophase of $T_b$ and activity) were respectively delayed or advanced relative to the onset of night under 22.5 h and 27 h

conditions (Fig. 1a, b; Supplementary Fig. 1). Thus, the high degree of phase coordination among individual mice and the long-term stability of their phase alignment relative to the LD cycle make the 22.5 h and 27 h LD cycle conditions ideal to test the physiological consequence of environmental desynchrony.

Energy metabolism is tightly coupled to the circadian clock[23–25]; therefore, the impact of 22.5 h and 27 h LD environment on body weight, adiposity, feeding behaviour and

energy expenditure was examined. In comparison to matched 24 h LD housed mice, a significant reduction in adiposity was observed in 22.5 h LD conditions (Fig. 2a). No significant differences were observed in comparable analyses of mice housed under 27 h LD conditions. Reduced fat mass in 22.5 h LD mice was exacerbated by12 months of 22.5 h LD exposure, by which time a significantly lower body weight was also evident (Fig. 2b). Consistent with reduced adiposity, 22.5 h LD housed

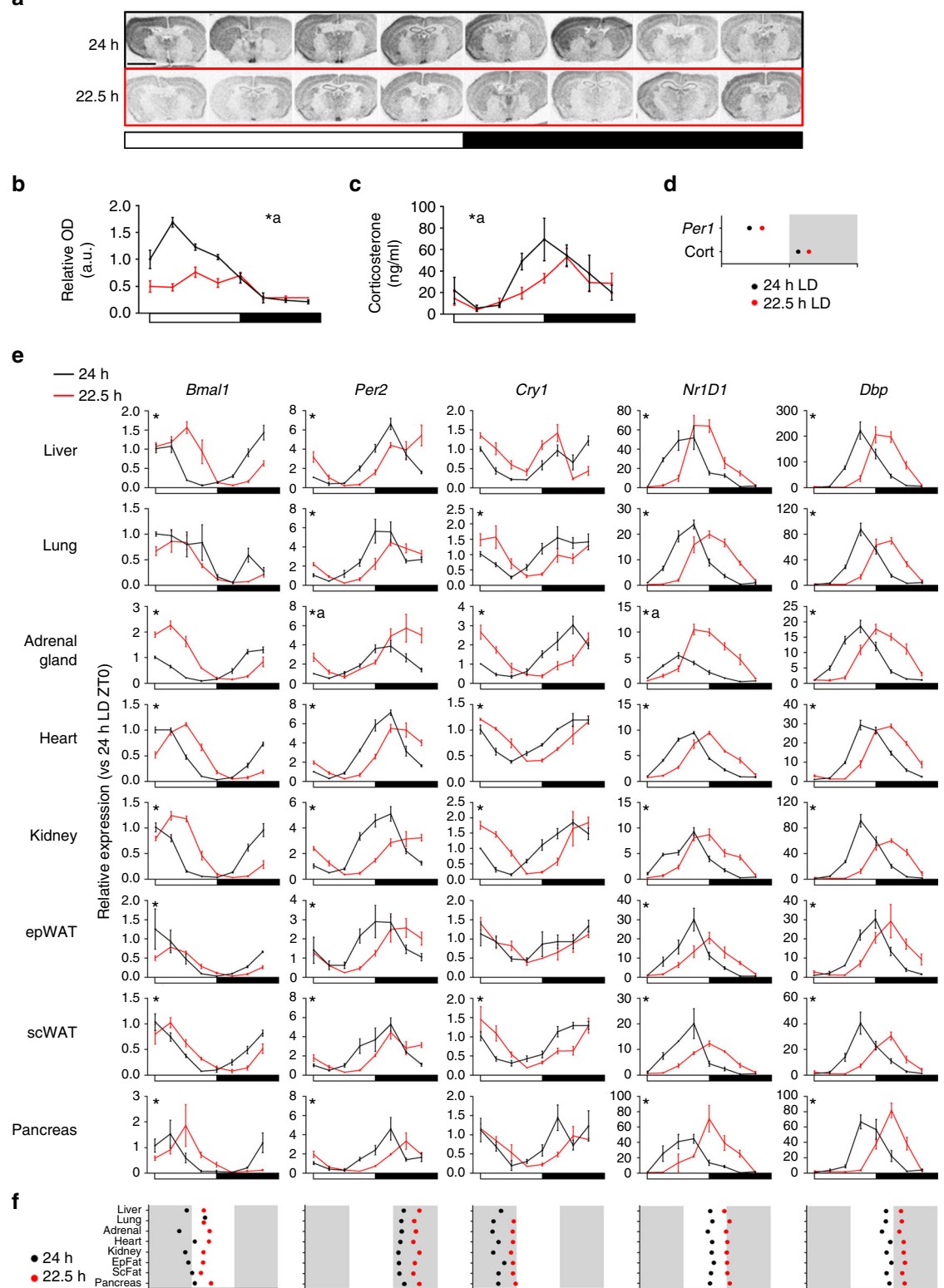

mice exhibited higher rates of both oxygen consumption (VO₂; Fig. 2c) and energy expenditure (EE; Fig. 2d). Interestingly, both the 22.5 h and 27 h LD conditions caused a significant increase in respiratory exchange rate (RER) and carbohydrate oxidation (COx) rate (Fig. 2e, f), indicating an increased reliance on carbohydrate substrates for energy generation in these mice. To assess feeding behaviour, mice were temporarily housed in automated feeding cages, and food intake monitored continuously for 5 days. In line with the increase in RER and COx, mice housed under non-24 h LD conditions exhibited a significant increase in food intake when compared with matched 24 h LD controls (Fig. 2g, h), which was due in large part to an increased intake during the light phase of the cycle (Fig. 2i). Together these findings show that entrainment to non-24 h LD cycles has a pronounced effect on energy efficiency (higher food intake without increased body weight) and substrate utilisation.

**Entrainment to non-24 h LD cycles does not disrupt clock function.** Despite the clear phase misalignment relative to the light cycle, mice entrained to non-resonant LD cycles exhibited a robust amplitude and consistent phase across different physiological parameters (Supplementary Fig. 1), suggesting that internal circadian timing is maintained in the animals. Therefore, we next sought to identify how central and peripheral oscillators respond to non-24 h LD cycles, with a focus on the 22.5 h LD condition. Quantification of *Per1* in the master SCN clock revealed a peak in expression during the early light phase within the 24 h LD housed mice (Fig. 3a, b). In line with the altered timing of activity and $T_b$ in 22.5 h LD housed mice, the rhythm in *Per1* expression in the SCN of these mice was significantly delayed in phase and dampened in amplitude relative to the 24 h LD group (Fig. 3a, b). This was reflected in circulating corticosterone, an important synchronising agent for peripheral tissue clocks whose secretion is strongly influenced by the SCN[26]. Corticosterone rhythms were significantly delayed in phase in mice entrained to 22.5 h LD cycles, when compared with 24 h LD housed animals (Fig. 3c, d). Rhythms in both SCN gene expression and corticosterone were significantly dampened in the 22.5 h LD housed mice, suggesting that the phase misalignment with the external LD cycle weakens the central oscillator.

Contrary to expectation, transcript profiles in peripheral tissues remained strongly rhythmic in both 24 h and 22.5 h LD housed groups (Fig. 3e). Strikingly, acrophase analyses revealed a consistent phase-delay relative to the LD cycle across all of the peripheral tissues examined (Fig. 3f), similar to that observed in the SCN. There was also a remarkable preservation of the phase relationship of clock components, both within and between tissue sets. A few exceptions were evident (e.g. *Bmal1* expression in the lung), but these were limited to single gene profiles in individual tissues. Thus, despite chronic entrainment to a short-running LD cycle, the circadian system remains intact and faithfully retains its amplitude and internal synchrony.

Many cellular processes including metabolism[27], cell cycle[28], oxidative stress[29], inflammation[30] and epigenetic modification[31] are subject to direct transcriptional control by components of the clock. Therefore, we next profiled regulatory factors and pathways known to be under circadian control (Fig. 4; Supplementary Fig. 2). The expression profile of many factors (e.g. *wee1*) matched that of the core clock, remaining robustly rhythmic but delayed in phase under 22.5 h LD conditions. However, many other factors (e.g. *nrf2*, *pparg*, *ezh2*, *RelA*) exhibited significant transcriptional dysregulation, with alterations in amplitude and/or mesor. Dysregulation of affected pathways (e.g. antioxident response, histone/DNA methylation) was highly tissue specific, demonstrating the tissue selective impact of phase misalignment (Supplementary Fig. 2). Thus, despite maintenance of robust core circadian clock function in non-resonant mice, disruption of timing was evident in a number of key clock-output pathways.

**Exposure to non-resonant LD cycles impairs cardiac function.** The heart houses a robust circadian clock (Fig. 3e)[32, 33], and cardiovascular health is adversely affected by shift work and forced desynchrony protocols[9, 34]. We therefore examined the impact of 22.5 h and 27 h LD cycles on heart rate (HR) and electrocardiogram (ECG) parameters. Mice exhibited a robust diurnal rhythm in HR under 24 h LD; however both 22.5 h and 27 h LD cycles disrupted the HR rhythms, and significantly slowed HR across the LD cycle (Fig. 5a; Supplementary Fig. 1d–f). The reduction in HR was not related to changes in locomotor activity (Fig. 5b; analyses limited to periods of >20 min of inactivity before HR measure). Critically, longitudinal analyses of ECG recordings in wild-type C57B6J mice revealed that non-resonant LD conditions not only increased inter-beat (RR) interval (Fig. 5d), but also slowed cardiac conduction parameters, with significantly lengthening of PR, QT and RR-adjusted QT (QTc) intervals evident in mice housed under either 22.5 or 27 h LD cycles (Fig. 5e–h).

Prolonged inter-beat intervals can result from changes in repolarising potassium channels[35]. However, we did not observe aberrant ion channel expression in the 22.5 h LD mice (Supplementary Fig. 2). In our studies, HR slowing occurred rapidly upon transition from 24 h to non-resonant LD cycles, and was evident in mice switched from 24 h LD to constant light (Supplementary Fig. 3). A similar LD cycle-dependent slowing of HR was observed in the accelerated period $CK1e^{tau}$ mutant mice (free-running period of 20 h), when housed under 22.5 h LD cycles (Supplementary Fig. 3). This suggests that aberrant light exposure drives altered cardiac function. Due to the altered phase of entrainment, both 22.5 and 27 h LD housed mice are exposed to light at inappropriate times (relative to their internal rhythm). Therefore, we assessed the effect of acute mistimed light exposure on HR by exposing control 24 h LD housed mice to a 2 h light pulse in the early active (dark) phase (ZT14-16). As expected, light exposure caused a significant reduction in activity, $T_b$ and HR. However, unlike $T_b$ and activity, mistimed light caused a profound decrease in HR that remained depressed in light-pulsed mice for a further two LD cycles as the mice regained normal phase alignment to the LD cycle (Fig. 5i, j; Supplementary Fig. 4).

**Fig. 3** Robust circadian rhythms in clock gene expression are maintained in non-24 h LD housed mice. Representative brain sections **a** and quantification **b** of radioactive in situ hybridisation for *Per1* expression in the SCN from mice housed in 24 h (*black*) or 22.5 h (*red*) conditions for 17 week. **c** Circulating corticosterone profiles were delayed relative to the LD cycle in mice housed under or 22.5 h conditions. **d** Acrophase analysis of *Per1* and corticosterone rhythms in 24 and 22.5 h housed mice reveals consistent phase delay. **e** Profiling of clock gene expression in peripheral tissues of mice housed in 24 or 22.5 h LD conditions for 17 weeks. **f** Acrophase analyses across genes and tissues (plotted relative to respective light cycle) demonstrates that synchronisation both within and across tissue clocks is maintained in non-24 h LD conditions. All data plotted mean ± SEM relative to the respective light cycle (24 or 22.5 h) and normalised to ZT0 of the 24 h LD group (*n* = 4/time-point/group). *Significant (*p* < 0.05) difference in phase between 24 and 22.5 h profiles; **a** significant difference in amplitude or mesor between 24 and 22.5 h LD conditions (sinusoidal waveform fits with F tests for shared characteristics). *Scale bar* in a = 3 mm

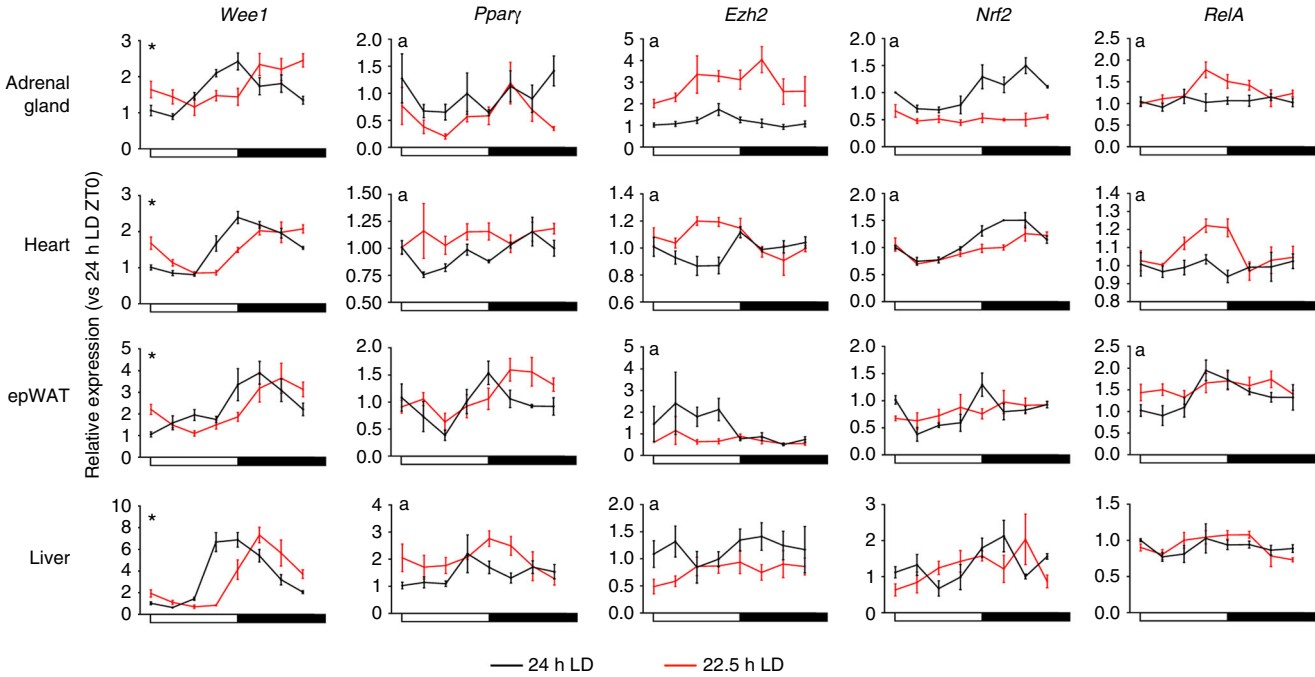

**Fig. 4** Phase misalignment disrupts rhythmic gene expression in a pathways and tissue-specific manner. Transcript profiles of major clock-controlled regulator genes of cell cycle (*Wee1*), epigenetic regulation (*Ezh2*), metabolism (*Pparγ*), oxidative stress (*Nrf2*) and inflammation (*RelA*) from mice housed in 24 or 22.5 h conditions for 17 weeks. All data plotted mean ± SEM relative to the respective light cycle (24 or 22.5 h) and normalised to ZT0 of the 24 h LD group ($n = 4$/time-point/group). *Significant ($p < 0.05$) difference in phase between 24 and 22.5 h profiles; a = significant difference in amplitude or mesor between 24 and 22.5 h LD conditions (sinusoidal waveform fits with F tests for shared characteristics)

HR can be acutely affected by environmental light via autonomic signalling[36, 37]. Our studies suggest that suppression of cardiac function and lengthening of QT, results from discordance between the phase of the local clockwork and extrinsic signals such as altered autonomic input to the heart. We tested this directly in 22.5 and 24 h LD housed mice by blocking vagal and sympathetic input to the heart using atropine/propranolol administration[38–40]. The impact of autonomic block on HR was assessed in both the active (2 h after lights on; ~ZT2) and inactive (2 h after lights off, ~ZT14) phases of the cycle under both lighting conditions. Indeed, at both time points, complete autonomic blockade caused a significant increase in HR under 22.5 h LD, but not in 24 h LD housed mice and resulted in comparable HR in both groups (Fig. 5k–m). Thus, our findings clearly demonstrate that discordance between environmental light/dark cycles and the internal clockwork cause a rapid and profound depression of cardiac function, including prolonged PR and QT interval. This effect is independent of locomotor activity, and is mediated by altered autonomic signalling.

## Discussion

These studies provide direct evidence that chronic desynchrony between the internal circadian clock and environmental light cycles profoundly impacts mammalian physiology. Housing mice under non-24 h LD cycles created an experimental paradigm, which mimics the phase misalignment experienced by humans with extreme chronotype and those engaged in shift work. Despite achieving stable entrainment, phase misalignment had a rapid impact on energy metabolism and cardiac rate and conduction, which did not diminish over many months. Remarkably, dysregulated metabolic and cardiac profiles were not driven by disruption of the molecular clockwork, nor by a loss of phase synchrony between different tissue oscillators (internal desynchrony). Instead, our studies clearly

indicate that it is the altered phase relationship between the external light cycle and internal circadian timing that drives the aberrant physiology.

Many animal studies have inferred the importance of circadian timing for maintaining a healthy physiological state through genetic disruption of one or more components of the underlying molecular clock. However, this approach rarely allows the importance of altered timing per se to be determined due to constitutive loss of the factor and its pleiotropic actions[19]. In contrast, resonance studies, as we use here, exploit the altered interaction between an intact biological clock and its external environment. The adaptive benefit of matching internal circadian time to environmental cycles has been robustly demonstrated in lower organisms[18], and a few studies have demonstrated that altered circadian timing can reduce survival fitness in natural settings[41, 42]. Yet few studies have used circadian resonance to reveal the impact of circadian timing to internal physiology and health in mammalian models. Here we show that under non-24 h LD cycles, mice exhibited markedly reduced energy efficiency and increased reliance on carbohydrate energy substrates. This finding reinforces the long held assertion that circadian timing serves to optimise response to the environment, including optimising cycles of energy storage and mobilisation. Decreased energy efficiency was at least in part due to increased energy expenditure in the 22.5 h LD housed mice. Phase misalignment may also lead to reduced nutrient absorption and/or utilisation as both are influenced by the circadian system[43]. In a notable study, Martino and colleagues reported that short period *CK1ε[tau/+]* mutant hamsters (endogenous period of ~22 h) were prone to cardio-renal dysfunction when housed under 24 h LD cycles, which did not manifest if the animals were maintained on a 22 h LD cycle[44]. In agreement with our studies, this indicates that an altered phase of entrainment is sufficient to drive pathology, and highlights a particular vulnerability of the cardiovascular system.

In pursuit of further consequences of environmental desynchrony, we profiled clock output genes across tissues. These studies showed tissue specific, and gene specific patterns of dysregulation, with enrichment for specific pathways (notably oxidative stress response in the heart and adrenal gland). Disruption of rhythmic transcription despite robust circadian clock gene oscillation has been demonstrated previously in both mice and humans[45–47], indicating a de-coupling possibly due to mistimed external signals. Moreover, it is

becoming clear that rhythmic processes within a tissue can be remodelled and detached from the circadian clock under environmental or pathological challenge[48, 49]. The altered expression of epigenetic regulators (e.g. EZH2, TET enzymes) in response to non-24 LD schedules, is strikingly similar to that observed in the SCN of 22 h LD housed mice reported by Brown and colleagues[45], and suggests that epigenetic mechanisms drive reprogramming of clock-controlled processes during phase misalignment.

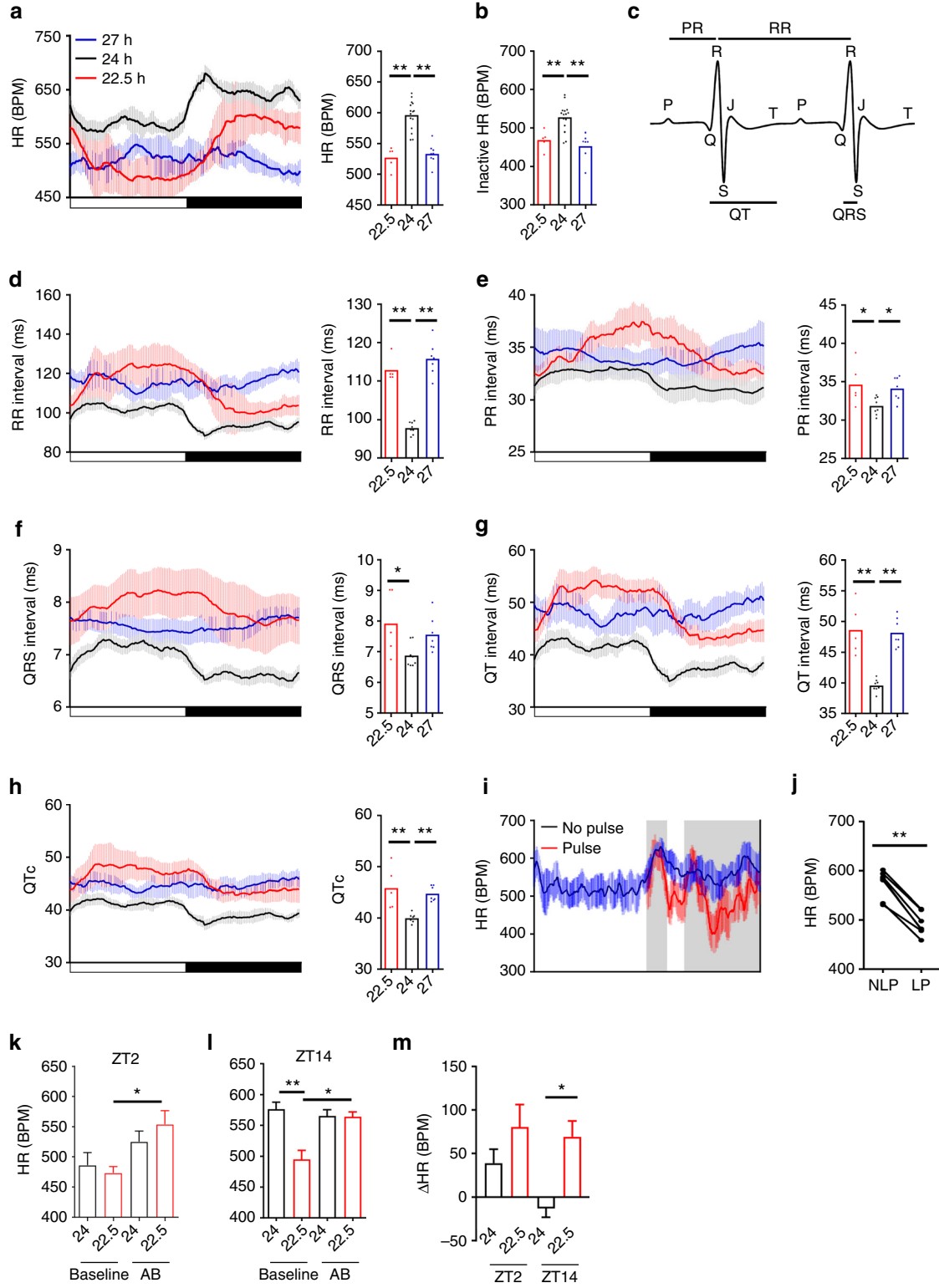

Cardiac physiology exhibits pronounced circadian rhythmicity, driven both by the local cardiac clock and through circadian variation in autonomic input to the heart[32, 33, 50–53]. In our studies, control mice exhibited robust rhythms in ECG parameters including RR, PR and QTc. All of these parameters were significantly prolonged when mice were placed under non-24 LD cycles. Altered cardiac conduction is associated with severe pathology and sudden cardiac death, and is therefore of profound interest to human health. The incidence of sudden cardiac death follows a strong diurnal rhythm, with peak incidence around the transition from sleep to wake[54, 55]; and increased incidence of heart disease and cardiac events is associated with nightshift workers[56]. Our results indicate that mistimed light drives inappropriate autonomic input to the heart, and suggests that circadian misalignment increases vulnerability to conduction block and ventricular arrhythmia (due to significant lengthening of PR and QTc). The dampening of the central SCN circadian oscillator observed in the non-resonant housed mice may weaken normal circadian control over autonomic output to the heart and other tissues, and thereby exacerbate phase misalignment between the local tissue clocks and autonomic drive.

Remarkably, depression of HR was observed in response to acute mistimed light exposure, which persisted over several days. The detrimental impact of mistimed light is clear, and is supported by reports of depressed HR and prolonged QT in mice housed under constant light or subject to light-restricted feeding[57]. Cardiomyocyte-specific deletion of Bmal1 leads to a similar lengthening of QRS and QTc intervals, which was ascribed to aberrant expression of cardiac ion channels Scn5a and Kcnh2[58, 59]. We did not observe altered expression of Scn5a, Kcnh2, or a number of other ion channels in cardiac tissue of 22.5 h housed mice. We show that autonomic block can normalise HR in non-resonant LD housed mice, demonstrating an underlying role for inappropriate autonomic signalling during phase misalignment with the LD cycle. In contrast to the mice, acute light exposure increases HR in humans. Nevertheless, important parallels exist between rodent and human HR/ECG responses to altered lighting/circadian desynchrony. Rats and humans exhibit a similar diurnal variation in the fractal structure of HR (a measure of HR variability), despite the opposite nature of diurnal/nocturnal activity in the species[60]. Most importantly, altered autonomic signalling and long QTc interval have been consistently reported in studies of shift work in humans (e.g. refs [61–63]). Thus, in both humans and mice, the strong influence of both extrinsic (autonomic) and intrinsic (local cardiac clock) timing signal to cardiac pacing is likely to increase the vulnerability of the cardiovascular system to dysfunction under conditions of circadian disruption.

By employing non-resonant light cycles, our studies isolate the physiological impact of circadian misalignment in a mammalian model. Given that the phase misalignment produced in our studies mimic the impact of extreme chronotype and shift work

on circadian phase in humans[22, 64–66], our findings provide important new insight into human pathologies associated with shift work, chronotype and social jetlag.

## Methods

**Animals.** All animal experiments were licenced under the Animals (Scientific Procedures) Act of 1986 (UK), and conducted in accordance with University of Manchester animal welfare committee guidelines. Male C57Bl/6J mice were purchased from Charles River (UK), and $Ck1e^{tau}$ mice[67] were bred at the University of Manchester. At 8 weeks of age, male mice were assigned randomly to each LD condition and transferred into light-tight housing cabinets and maintained in 20, 22.5, 24, 27 or 30 h LD cycles for 17–52 weeks. Light levels were maintained at 463 lux during the light phase and during light pulse experiments. Ambient temperature was $22 \pm 2\,^{\circ}C$, with food and water available ad libitum. Mice remained group housed throughout, except where single housing was necessitated by physiological monitoring (e.g. CLAMS, ECG telemetry, food intake recording). Due to different lighting schedules, blinding of the experimenter to experimental conditions was not possible.

**Behavioural and physiological monitoring.** To assess body composition, whole body lean and fat mass was assessed using the EchoMRI system (Echo Medical Systems). For long-term (>10 week) recording of body temperature, mice were implanted with iButton temperature loggers (Maxim, DS1922L-F5). Implants were de-housed, programmed and encapsulated in a 20% Poly(ethylene-co-vinyl acetate) and 80% paraffin mixture[68] before implantation into the peritoneal cavity (ip). Temperature recordings were calibrated by immersion of the iButtons into set temperature water baths before implantation. In separate experiments, $T_b$ and locomotor activity rhythms were recorded using indwelling radio-telemetry devices (TA-F10, DSI international) implanted ip. Continuous measures of food intake were recorded from singly housed animals using PhenoMaster behavioural cages (TSE Systems). Mice (>8 week LD cycle exposure) were acclimatised to the cages for two cycles, after which food consumption was measured in 5 min bins for 3–4 LD cycles. To assess metabolic gas exchange, mice were individually housed in indirect calorimetry cages (CLAMS, Columbus Instruments). As above, mice (>8 week LD cycle exposure) were acclimatised to the cages for two cycles, following which $O_2$ consumption and $CO_2$ production were recorded every 10 min for >3 LD cycles. RER was derived from these measures ($VCO_2/VO_2$), as were protein oxidation independent COx ($4.55^*VCO_2 - 3.21^*VO_2$)[69] and energy expenditure ($3.815^*VO_2 + 1.232^*VCO_2$).

**Heart rate and ECG recording.** Mice were implanted with ETA-F10 radio-telemetry devices (Data sciences international) for recording of locomotor activity, $T_b$ and ECG. Devices were implanted ip with ECG recording leads brought through the abdominal wall and negative lead secured ~1 cm right of midline at upper chest, positive lead secured ~1 cm to the left of midline at the xiphoid plexus. Mice recovered for 7–10 days before the start of recording the experiment. Mice were excluded from the study if lead placement was not maintained throughout the experiment. For longitudinal studies (i.e. >7 days), activity, $T_b$ and 10 s ECG data sweeps were collected every 10 min, and a minimum of three consecutive days used for each analyses period. 'Inactive heart rate' was defined as periods where no activity had been recorded for >20 min before the ECG sweep recording. Following recording, ECG waveforms (10 s sweeps at 1 sweep/10 min) were processed and analysed using bespoke software programme written in Matlab (Mathworks). In brief, ECG R waves (local maxima) were extracted by amplitude windowing (typically 0.3–3 V) and subsequent template matching to the mean beat waveform. For this analysis, we calculated an ECG waveform template for each individual sweep (the normalised mean of all beats within a pre-defined amplitude window) which then served to identify and remove occasional spurious threshold crossings due to noise. Individual recording sweeps were excluded from subsequent analysis where the mean amplitude of discriminated beat waveforms was < 3 times the lower limit of the amplitude window, where baseline variation in the ECG trace exceed 1/3 of that lower limit and/or where > 20% of the events detected were excluded by the template matching algorithm. Specific ECG parameters (RR, PR,

**Fig. 5** Misalignment with environmental LD cycles disrupts cardiac rate and leads to long QT interval. **a** Incomparison to matched 24 h LD cycle housed mice, mean heart rate (HR) was significantly reduced in mice maintained in 22.5 and 27 h LD cycles, which was accompanied by a significant reduction in the amplitude of HR rhythms in the 27 h LD mice. **b** HR remained significantly reduced in non-24 h LD housed mice even when analyses were limited to periods of inactive (no recorded movement for 20 min preceding HR measure). **c** Murine ECG beat waveform with defining features labelled. **d–h** Diurnal profiles and mean interval durations for RR (**d**), PR (**e**), QRS (**f**), QT (**g**) and RR-adjusted QT (QTc; **h**) of mice maintained under 24 h, 22.5 h or 27 h LD conditions. All parameters were significantly lengthened under non-24 h LD cycles. **i, j** A 2 h light pulse (LP, red) from ZT14-16 caused a significant reduction in night-time HR when compared with the average HR profile recorded over three preceding cycles (blue, NLP). **k–m** Complete autonomic blockade (AB) increased HR at ZT2 (~2 h after lights on; **k**) and ZT14 (~2 h after lights off; **l**) in 22.5 h LD housed mice, and normalised it to 24 h LD conditions. **m** Change in HR (vs. baseline) during autonomic blockade at ZT2 and ZT14. Diurnal profiles depict mean ± 95%CI. *$p < 0.05$, **$p < 0.01$. **a–h** One-way ANOVA with Dunnett's post hoc test (vs. 24 h control group); **j** paired t-test; **k–m** repeated measures two-way ANOVA with Tukey's post hoc test

QRS, QT) were then collected for each beat by automated identification and recording of appropriate deflection points between each valid R–R interval in the trace. Within our data sets >80% of analysed traces passed this quality control step and, as a matter of routine, we manually inspected randomly chosen subsets of valid and excluded traces to confirm appropriate categorisation and analysis. Importantly, we found that using more or less stringent quality control parameters did not affect the overall results of our subsequent analysis, nor did we find any bias with respect to the proportion of traces excluded from analysis during active vs. inactive portions of the daily cycle. For autonomic blockade, conscious free-moving 22.5 h and 24 h LD housed mice were injected either 2 h after lights on (~ZT2) or 2 h after lights off (~ZT14) with atropine (0.5 mg/kg, ip) followed by propranolol (1 mg/kg, ip). During autonomic blockade studies, radio-telemetry data (activity, $T_b$ and HR) was collected every minute. Baseline HR was derived from 1 h pre-injection, with HR during complete autonomic blockade collected 20–30 min post-administration. Injection-induced increases in HR following vehicle administration (conducted 1 cycle before autonomic blockade) returned to baseline by 15 min post-injection.

**Gene expression analyses**. For qPCR, tissues were rapidly dissected and snap frozen at 8 time-points spaced equally through the 24 and 22.5 h LD cycles. Total RNA was extracted using Trizol reagent (Life Technologies) according with the manufacturer's protocol. cDNA was synthesised using High capacity RNA to cDNA kit (Applied Biosystems). qPCR was performed using GoTaq Master Mix (Promega) and an applied biosystems 7900 384 well thermal cycler (Applied Biosystems). Relative gene expression was quantified using the ΔΔCT method using Ppib and r18s as reference genes. Primer sequences are listed in Supplementary Table 1. For in situ hybridisation, brains were frozen over dry ice, cryo-sectioned (12 μm), mounted onto poly-L slides, and stored at −80 °C. Per1 plasmid was a kind gift from Prof Urs Albrecht (University of Freibourg). The cDNA fragment was cloned into a pGEMT-easy vector (Promega) and linearised using BamHI and XhoI to produce sense and antisense templates, respectively. Riboprobes were synthesised in the presence of 33UTP, and hybridised overnight at 60 °C. Hybridisation was visualised by exposing autoradiographic film at −80 °C for 7 days. Signal intensity was quantified by densitometry analysis of SCN region normalised to background staining. Normalised optical density for the SCN reflects >3 sections per animal and 3–4 animals/tp.

**Statistical analyses**. Experimental design and *n* number determination was based on previous experience and appropriate power analyses ($\alpha = 0.05$, 80% power, 20–50% estimated effect size depending on measure). Data are presented as mean ± SEM or 95% CI as indicated. Mean difference comparisons were carried out using Student's *t*-test (two-sided; paired where appropriate), one-way ANOVA, and two-way ANOVA with post hoc tests and repeated measures as appropriate. Determination of acrophase in clock gene expression (Fig. 3) was performed by harmonic regression using CircWave v1.4 software[70]. For statistical comparison of rhythmic characteristics (mesor, phase, amplitude) of gene expression and corticosterone profiles, best-fit sinusoidal waveforms were generated by regression analyses and shared characteristics tested with equal sum of square F tests (Graphpad Prism 7.0). For longitudinal measures (e.g. HR), sinusoidal waveforms were generated for each individual, from which rhythm characteristics were derived and group mean differences tested with one-way ANOVA.

**Data availability**. Materials, algorithms and data generated in these studies are available from the authors upon reasonable request.

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

## Acknowledgements

We thank the Biological Services Unit at the University of Manchester for support of in vivo studies. This work was supported by the Biotechnology and Biological Sciences Research Council (UK) through grants to D.A.B. (BB/I01864/1; BB/J017744/1) and T.M.B. (BB/N007115/1).

## Author contributions

All authors contributed to experimental design; A.C.W., L.S. and D.A.B. conducted experiments; A.C.W., T.M.B. and D.A.B. analyzed results; A.C.W., D.W.R., A.S.I.L., T.M. B. and D.A.B. wrote the manuscript; D.A.B. conceived the project and supervised all aspects.

## Additional information

**Competing interests:** The authors declare no competing financial interests.

