## [Peer Review file · Nature Communications]

Reviewers' comments:

Reviewer #1 (Remarks to the Author):

West et al. determined the effect of exposure to non-24 h light/dark (LD) cycles on rhythms in behaviors, and physiologic, metabolic and cardiac function as well as gene expression in various organs in mice to study the disruptive effects of circadian misalignment during stable entrainment on metabolic and cardiac function. They show that exposure to 22.5-h and 27-h LD cycles, but not to 20-h and 30-h LD cycles, leads to stable entrainment of the temperature rhythm, but with a changed phase relative to the LD cycle. They next show that, under these conditions of misalignment, metabolic efficiency is decreased and cardiac function changes, including a reduction in the QTc. These changes are observed despite core clock gene expression remaining robust in peripheral tissues and without changes in the timing clock gene expression rhythms between peripheral organs, including adrenal cortex, liver, and heart, indicating that misalignment of behavioral, physiological, and molecular rhythms relative to the LD cycle, while stably entrained, leads to important cardiac and metabolic changes. This study addresses an important research question, the studies are well designed and executed, the assessments are comprehensive, the manuscript is well written, and the results may have direct relevance for extreme chronotypes who typically live entrained but at abnormal phase angle with the LD cycle. However, there are a number of concerns to be addressed.

- In the Abstract it is concluded "Importantly, physiological dysfunction was not driven by disrupted core clock function, nor by an internal desynchrony between organs, but rather the altered phase relationship between a robust internal clockwork and the external environment." However, while the core clock gene expression rhythms in peripheral organs remained robust, the core clock function in the SCN, an integral part of the circadian system, was severely blunted, i.e., disrupted and not robust. Also the corticosterone rhythm, a major output of the circadian system, was significantly blunted. Therefore these conclusions, in the Abstract and throughout the manuscript, need to be corrected accordingly. Related to this, discuss the possible contribution of these amplitude reductions of the SCN and corticosterone rhythm in the observed transcriptional, metabolic and cardiac changes.

- There are a number of 'first-time' claims that are described too broadly, making them difficult to justify. For example, in the Abstract, it is concluded "We reveal for the first time that even under stable LD environments, phase desynchrony has a profound effect..." And in the Discussion (line 228) "By employing non-resonant light cycles, our studies isolate for the first time the physiological impact of circadian misalignment in a mammalian model.". However, as example, Salgado-Delgado et al. (e.g., PMID: 20080873) and Bray et al. (PMID: 22907695) have shown that under a stable LD cycle, misalignment of the rest/activity cycle and/or the feeding/fasting cycle relative to the LD cycle results in profound effects on metabolism, and Martino et al. (e.g., PMID: 18272659), Scheer et al. (PMID: 19255424) and Wright et al., (e.g., PMID: 25640603) have shown that stable LD cycles that do not match their endogenous circadian period leads to pronounced physiological disturbances. Either remove the first-time statements or make them more specific.

- The stable entrainment with an abnormal phase angle of behavioral and physiological rhythms relative to the light/dark cycle under the 22.5-h and 27-h LD cycles is imprecisely referred to as desynchrony. In the strict term, desynchrony refers to a state where there is no entrainment or synchrony, i.e., where the period of two oscillations differs. The abnormal phase angle during the 22.5-h and 27-h LD cycles could be referred to as delayed or advanced phase, respectively, or misalignment, which refer to properties of phase, not period.

- It is shown that the 22.5-h and 27-h LD cycles decrease energy efficiency (increased food intake without increased body weight). Was there an increase in Tb that could help explain this? Have the authors measured nutrient absorption, e.g., by bomb calorimetry? Please discuss these possible mechanisms to explain a decreased energy efficiency.

- It is concluded that non-resonant LD cycles do not cause disruption or desynchrony within central or peripheral tissue clocks (line 93). Have the authors assessed the influence of the 20-h and 30-h LD cycles, also non-resonant LD cycles (albeit non-synchronizing), on SCN and

peripheral tissue clocks?

- Line 106: "Rhythms in both SCN gene expression and corticosterone were significantly dampened in the 22.5h LD housed mice, suggesting that the phase misalignment with the external LD cycle weakens the central oscillator." A dampening of the rhythm in group data could be due to two different effects. It can be caused by a dampening of the amplitude within each individual animal, or it can be caused by a larger phase dispersion between individual animals causing a blunting of the averaged group rhythm. While the in-situ data cannot distinguish this (no repeated measurements within an animal), data on cortisol and Tb enable testing these different scenarios. Show data on the individual phase dispersion and discuss this distinction.

- The effect of the 22.5-h LD cycle on approximately 16 transcripts (in addition to the core clock genes) are presented in peripheral tissues. Were there any other transcripts tested besides the ones presented? How were the specific transcripts selected, as opposed to other candidates that could have been selected for the pathways of interest?

- How is the lower HR explained despite a higher metabolic rate? E.g., was there a change in heart size suggesting increased cardiac output?

- Line 129, 134, and 154: check that the correct Figures and panels are referred to.

- Line 186: "... circadian timing serves to maximise response to the environment..." A larger response is not per se a sign of a well-adapted organism. Consider replacing "maximize" with, e.g., "optimize".

- Line 188: Replacing "Ralph and colleagues" with "Martino and colleagues".

- In paragraph 204-214, the effects of light on cardiac control is translated to human health. In this context it is important to discuss that effects of light in humans and mice can be opposite, i.e., a decrease in HR in mice but an increase in HR in humans (e.g., PMID: 10452332).

- Line 265: Was activity and Tb only assessed for 10 sec every 10 min? Does this mean that an animal could have been active for, e.g., 9 min while the activity recording shows no activity? Was the 10-sec ECG sweep done every 10 min (line 265) or every 5 min (line 269)?

- Provide more details on the software, version, and templates used for the ECG analysis. Also, was analysis done fully automated, manually, or mixed?

- Line 282: Was injection done i.p., i.v., or otherwise? Was there a vehicle control to control for the effect of the injection itself on stress-induced changes in HR? If not, justify why this was not necessary. What was the time of injection in the different groups?

- What were the light intensities in the study during L and D, especially in the light exposure studies?

- In panels "a" of Fig. 1, what is the top-to-bottom range in Tb of each line and with what resolution? Is the top-to-bottom range the same for each condition (the five columns) and each day (each line top to bottom)?

- Fig. 2, it is indicated that there were more than 13 animals in each group. It seems there are 7 data points for 27h epWAT. Even though one or two data points may be 'hidden' behind each other, it is unlikely that there are 14 data points. Please clarify.

- Fig. 5: Provide statistics for the statement in the Legend: "Mean heart rate (HR) and diurnal rhythms in HR were significantly reduced ...", and clarify what is meant with "reduced... rhythms", presumably "reduced rhythm amplitudes". Change the first "d" in the Legend to "c". Why is the profile for RR much more smooth compared to that for HR? Is it based on different data sets?

Panel i: which study is this from, how long were the animals exposed to LL, and explain the study rationale. "I. HR recorded during the LP was also significantly lower than that recorded the previous light (L) or dark (D) phase." This conclusion is not clear from the data displayed in Fig. S3f. Please explain why these results seem different.

- Fig. S3: What is meant with "transition period"? Transition from what? Also correct this second sentence that seems to be incomplete. When are days 4-10 in the figure? HR is substantially lower the days after the L pulse. How are Tb and activity affected? What is the proposed mechanism that causes this after-effect of the L pulse?

Reviewer #2 (Remarks to the Author):

The concept that disturbance of the physiological 12h light cycle in mice leads to physiological disturbance is not new. Thus, the study is of moderate interest and remains merely descriptive. For example, Karatsoreos et al. found that chronic circadian disruption by housing mice in 20-h light/dark cycles results in accelerated weight gain and obesity, as well as changes in metabolic hormones. Data from Martino et al (Hypertension 2007) show that circadian disruption promotes organ dysfunction in a pressure overload-induced mouse model of cardiac remodeling. They also demonstrate that disturbing diurnal rhythms also affects the clock genes *Per2* and *bmal* in the SCN, whereas rhythmic expression of these clock genes remains robust in the heart. Alibhai et al (Circ Res 2014) reported that rhythm disruption affects body metabolism (e.g. whole body substrate use, respiratory exchange ratio) and myocardial infarction healing.

Minor comment:

Please check your reference list, since ref. 33 and 53 are listed twice.

Reviewer #3 (Remarks to the Author):

Numerous studies have suggested links between disturbed biological rhythms and health. This is of profound concern to public health given the myriad ways in which modern society forces us to work at odds with our circadian system, be it via shift work, jet lag or nocturnal light exposure. However, much of the data of the health effects of circadian desynchrony is correlative, and the mechanisms underlying effects on physiology are poorly understood. Genetically modified animals carrying clock gene mutations have been shown to exhibit a range of different health problems, but a major issue with these studies is that developmental effects or the role of these genes in other processes could account for these changes. Moreover, as the molecular circadian clock regulates patterns of gene expression in a tissue-specific manner, disturbances in these normal transcriptional profiles could also give rise to such changes. It has often been suggested that the negative health consequences of circadian disturbances arise due to internal desynchrony – that the endogenous clocks in different organ systems become de-synchronised/misaligned. However, others have suggested that it is the mismatch between internal and external time that underlie the pathologies associated with circadian disturbances.

This manuscript directly addresses this issue by studying wildtype mice under different environmental conditions. Rather than studying mice with aberrant circadian clocks living under 24h light/dark (LD) cycles, instead the authors use healthy wildtype mice housed under non-24h LD cycles. This 'circadian resonance' approach has been elegantly applied to the study of non-vertebrate clocks, but has not previously been applied to mammalian circadian biology in this manner. As such, this provides a powerful approach to study the effect of desynchrony between the external environment and the internal biological clock. The authors initially study 20h, 22.5h, 24h, 27h and 30h LD conditions, showing that under 22.5h and 27h mice still entrain, but with a delayed or advanced phase. Under 20h or 30h conditions, mice free-run, adopting a dynamically changing relationship between internal and external time. The authors go on to study both metabolic and cardiovascular parameters under these conditions, showing that 22.5h LD conditions exert significant effects on metabolic and cardiovascular physiology. Moreover, they show that clock gene expression throughout the body is retained under these conditions, albeit with a different phasing. Finally, they show that mistimed light exposure exerts quite dramatic effects on heart rate, which appear to be mediated via the autonomic nervous system.

Overall, this is a very elegant study which has been well-designed and executed. The data are both extensive and provide a convincing case that the misalignment between internal circadian time and the external light environment underlies key changes in mammalian physiology which have clear implications for human health and disease.

MAIN COMMENTS

1. How was rhythm amplitude affected by these different conditions? Under T20 and T30, rhythms are clearly attenuated, but were any changes in amplitude, light phase activity etc; also observed?

2. How do the authors reconcile their findings of decreased adiposity with the view that circadian disturbances give rise to increased weight gain and metabolic disease? These findings are not inconsistent – the change in food intake and reliance on carbohydrate substrates may produce changes in behaviour giving rise to different effects if alternative food sources were available (e.g. high fat diet). These differences should be discussed.

3. In figure 3, 4 and S2, how were differences in phase determined? The CircWave software citation refers to an unpublished methodology, so it is unclear how differences between phase and amplitude under the different conditions were determined.

MINOR COMMENTS

1. In Figure 5j the blue line (no pulse) shows a clear response to the light pulse, whereas the red (pulse) does not. I presume this is mislabelled?

2. Line 152-154 refers to Figure S4, which is the CK1 ϵ data. I presume this should be Fig 3?

Reviewer #1 (Remarks to the Author):

West et al. ...important research question, the studies are well designed and executed, the assessments are comprehensive, the manuscript is well written, and the results may have direct relevance for extreme chronotypes who typically live entrained but at abnormal phase angle with the LD cycle.

We appreciate the strong support provided by the reviewer.

1. While the core clock gene expression rhythms in peripheral organs remained robust, the core clock function in the SCN, an integral part of the circadian system, was severely blunted, i.e., disrupted and not robust. Also the corticosterone rhythm, a major output of the circadian system, was significantly blunted. Therefore these conclusions, in the Abstract and throughout the manuscript, need to be corrected accordingly...discuss the possible contribution of these amplitude reductions of the SCN and corticosterone rhythm in the observed transcriptional, metabolic and cardiac changes.

We accept the comments of the reviewer, and indeed the weakened SCN and corticosterone rhythms were highlighted in the original manuscript (lines 100-107). We have now further amended the text as suggested by the reviewer (lines 20 and 173), and have added text to the discussion (lines 219-222) to address the potential contribution of weakened SCN rhythms on the physiological impact of non-resonant LD.

2. There are a number of 'first-time' claims that are described too broadly, making them difficult to justify.... Either remove the first-time statements or make them more specific.

We acknowledge that some of these statements were defined too broadly, and have therefore removed the 'first-time' qualifier from lines 16, 55, and 241. However, the majority of the studies highlighted by the reviewer (and others in the literature) employ a 'forced routine' in order to drive desynchrony between behaviour and the internal clock. For example, restricting food availability or forced activity during the normal rest period in animal model, and use of a prescribed constant routine in human studies. These models are valuable for desynchronising behavioural cycles from the internal clock (and/or LD cycle), but are distinct from the chronic phase misalignment examined here. Notable exceptions of course include work by Martino et al on the tau mutant hamster, as well as some studies by Ken Wright and colleagues in humans (which are cited in our manuscript).

3. ...desynchrony refers to a state where there is no entrainment or synchrony, i.e., where the period of two oscillations differs. The abnormal phase angle during the 22.5-h and 27-h LD cycles could be referred to as delayed or advanced phase, respectively, or misalignment, which refers to properties of phase, not period.

Discordance between the oscillating frequencies of the external LD cycle and the internal clockwork of the animals is a fundamental basis of the resonance testing. Nevertheless, we accept that in some instances 'phase misalignment' would be more appropriate to characterise the physiological response to 22.5 and 27 hr LD cycles. We have therefore replaced 'desynchrony' where appropriate in the manuscript text, including in the title (lines 1, 17, 49, 171, and 231).

4. It is shown that the 22.5-h and 27-h LD cycles decrease energy efficiency (increased food intake without increased body weight). Was there an increase in Tb that could help explain this? Have the authors measured nutrient absorption, e.g., by bomb calorimetry? Please discuss these possible mechanisms to explain decreased energy efficiency.

Both the 22.5h and 27h LD housed mice were found to have increased food intake, but not increased

Response figure 1. Increased active phase 'alpha' in 27h LD housed mice. The proportion of the LD cycle in which the animals maintain a body temperature (Tb) above their cycle mean was decreased and increased in the 22.5h and 27h LD housed animals, respectively. * P<0.01 versus 24h LD One-Way ANOVA.

body weight. As we showed in the original manuscript, we did not observe any group differences in mean body temperature between the LD groups (Supplementary Figure 1). However, the 22.5h LD housed animals exhibited an increase in metabolic rate (Figure 2), which will have certainly contributed to the reduced weight gain. In mice, use of body temperature as an indication of energy expenditure must be approached with caution. When housed below thermoneutrality (28-30°C), mice actively defend body temperature. Because of this, increased energy expenditure and metabolic rate may occur with little notable change in body temperature. Increased oxygen consumption was not observed in the 27h LD housed mice (in which the body weight and adiposity phenotype was less pronounced). Changes in the activity pattern of the mice housed under 27h LD may have contributed to an altered energy efficiency in the animals. For example, body temperature rhythms exhibited an increase in *alpha* duration in the 27h LD housed mice when compared with those housed under 24h (Figure 1, Response Figure 1). This may have contributed a subtle, yet significant effect on energy efficiency over the 16 weeks of altered LD exposure.

We did not assess faecal nutrient content nor nutrient absorption in the animals. Given the chronic nature of our studies (17 weeks to 1 yr), these studies represent a major undertaking, which we believe is beyond the scope of this current manuscript. We have added text to the discussion to address this point (line 189-192).

5. It is concluded that non-resonant LD cycles do not cause disruption or desynchrony within central or peripheral tissue clocks (line 93). Have the authors assessed the influence of the 20-h and 30-h LD cycles, also non-resonant LD cycles (albeit non-synchronizing), on SCN and peripheral tissue clocks?

We have not analysed clock gene expression in the mice housed under 20h or 30 h LD cycles. Due to its relevance for extreme chronotype/desynchrony in humans, our studies have been focused specifically on the non-resonant LD cycles to which the mice are able to stably entrain. The 20 and 30h LD models are greatly complicated by the fact that the mice come in and out of phase with the light cycle (thereby creating a variable impact of light-induced masking and phase shifts). Because of this, we do not feel that the large number of mice required for a cross-sectional tissue collection across 20h and 30h LD cycles is justified. To address the reviewers concerns, we have modified the results section subheading to specify 'Entrainment to non-resonant LD conditions do not...' (lines 93). As observed in Figure 1 and new analyses added to Supplementary Figure 1, animals under the 20h and 30h LD cycles are able to maintain robust rhythms in body temperature, suggesting that the SCN clock is relatively robust within these mice. However, body temperature profiles do reveal increased phase dispersion between individuals within the 20 and 30h LD groups, and decreased temperature rhythm amplitude in the 20h LD housed mice (Supplementary Figure 1).

6. Line 106: "Rhythms in both SCN gene expression and corticosterone were significantly dampened in the 22.5h LD housed mice, suggesting that the phase misalignment with the external LD cycle weakens the central oscillator." A dampening of the rhythm in group data could be due to two different effects. It can be caused by a dampening of the amplitude within each individual animal, or it can be caused by a larger phase dispersion between individual animals causing a blunting of the averaged group rhythm. While the in-situ data cannot distinguish this (no repeated measurements within an animal), data on cortisol and Tb enable testing these different scenario's. Show data on the individual phase dispersion and discuss this distinction.

The reviewer raises an important point. This data was presented in our original manuscript in Supplementary Figure 1a, which showed acrophase analyses of body temperature, VO₂, HR, and activity rhythms for the 22.5h, 27h, and 24h LD cycle housed mice. For additional clarity, we have now included additional analyses into Supplementary Figure 1, which illustrate the individual animal Tb and HR rhythms waveforms (derived from regression analyses). The phase dispersion among individual mice housed under 22.5h, 24h, or 27h LD cycles is relatively small. Therefore, damping of amplitude within the cross-sectional sampling studies (e.g. gene expression in SCN and peripheral tissues) is not likely to be greatly affected by phase differences between individuals. Corticosterone rhythms were not derived from serial bleeds for each individual, but from terminal blood collection

(same studies which provided *in situ* and peripheral tissue qPCR profiles). Thus, we cannot assess individual mouse corticosterone profiles. The marked phase separation observed with the 20hr and 30 hr groups further strengthens our argument to focus on the 22.5h, and 27h groups.

7. The effect of the 22.5-h LD cycle on approximately 16 transcripts (in addition to the core clock genes) are presented in peripheral tissues. Were there any other transcripts tested besides the ones presented? How were the specific transcripts selected, as opposed to other candidates that could have been selected for the pathways of interest?

We chose the transcripts shown in Figure 4 and Supplementary Figure 2, due to their roles as key regulators in the paths of interest, their known connection to the circadian clock, and their physiological relevance in multiple tissues. We profiled approximately ~60 transcripts in various tissues, and the set of gene profiles included in the manuscript reflects the general transcriptional response that we observed in the non-resonant mice. As stated in the manuscript text, many gene profiles were disrupted by the non-resonant LD cycle; however, these effects were both gene and tissue specific. This is evident in the pathways highlighted in the original manuscript (Figure 4 and Supplementary Figure 2). We acknowledge that this level of coverage cannot provide a non-biased genome wide representation of transcriptional response to non-24h LD. However, we believe that they do reflect the selective yet significant impact of the phase misalignment on transcriptional control in the mice. Altered expression of DNA and histone methylation machinery provides a mechanism for such an impact. Unfortunately, we do not have sufficient resources available to attempt genome wide RNAseq analyses on such a complex experimental paradigm (multiple tissues, time-points, and LD conditions).

8. How is the lower HR explained despite a higher metabolic rate? E.g., was there a change in heart size suggesting increased cardiac output?

We examined heart size in two studies under 22.5h LD cycles; however, we did not observe a consistent effect of the non-resonant LD cycle. Specifically, after 17 weeks of non-resonant LD exposure, heart weight (relative to body mass) was increased in 22.5h LD housed mice when compared with 24h housed mice (Response Figure 2), supporting the reviewer's suggestion. However, by 1 year of altered LD exposure, heart weight (relative to body weight) and cardiomyocyte size was reduced in the 22.5h housed mice compared to matched 24hr housed animals. It is possible that the long-term housing under non-resonant LD, and the resultant reduction in heart rate eventually leads to reduction in cardiac tissue mass. Nevertheless, due to the relatively crude measures at 17 wk (dissected heart weight) and differences observed between early (17 wk) and long-term (52 wk) exposure, we do not feel that it is appropriate to include this data in the current manuscript.

Response figure 2. Impact of non-24h LD on heart weight. A) 22.5h LD exposure for 17 weeks caused a small but significant increase in heart weight, compared with 24h LD housed mice (n = 32/group). **B)** By 52 weeks, heart weight was lower in the short LD housed mice, with significant reduction in ventricle wall thickness and cardiomyocyte size (n = 5/group); *P<0.05 versus 24h LD, Student's T-Test

9. Line 129, 134, and 154: check that the correct Figures and panels are referred to.

128 changed to Fig 3e

133 is Fig 5b

152 This has been corrected, although supplementary Figures 3 and 4 have been modified.

10. Line 186: "... circadian timing serves to maximise response to the environment..." A larger response is not per se a sign of a well-adapted organism. Consider replacing "maximize" with, e.g., "optimize".

This has been amended (188).

11. Line 188: Replacing "Ralph and colleagues" with "Martino and colleagues".

This has been amended (192).

12. In paragraph 204-214, the effects of light on cardiac control is translated to human health. In this context it is important to discuss that effects of light in humans and mice can be opposite, i.e., a decrease in HR in mice but an increase in HR in humans (e.g., PMID: 10452332).

We have now added extra discussion to this section (lines 231-36) acknowledging the similarities and differences between light-induced HR responses in mice and humans. As noted by the reviewer, acute light exposure leads to a stimulatory effect in humans (including increase in HR), whereas it has a suppressive effect in rats and mice. This is in keeping with the respective diurnal/nocturnal activity patterns. Nevertheless, it is the disturbance of normal HR control by altered lighting which is likely to have adverse health consequence, rather than acute increases/decreases in HR. In this regard, important parallels exist between rodent and human HR/ECG responses to altered lighting/circadian desynchrony. Most importantly, autonomic imbalance and long QT in response to shift work is consistently reported in human studies and experimental desynchrony (Mozos and Filimon 2013; Lee et al 2015; Meloni et al 2013; Yoshizaki et al 2013, Ishii et al 2004, 2005; Holmes et al 2001, and many others). Moreover, rats and humans exhibit a similar diurnal variation in the fractal structure of heart rate (a measure of HR variability), despite the opposite nature of

diurnal/nocturnal activity in the species (e.g. Hu et al 2008). Altered autonomic balance appears to be a major pathological consequence in both rodents and humans, and thus our results provide important insights for human health.

13. Line 265: Was activity and Tb only assessed for 10 sec every 10 min? Does this mean that an animal could have been active for, e.g., 9 min while the activity recording shows no activity? Was the 10-sec ECG sweep done every 10 min (line 265) or every 5 min (line 269)?

We apologise for the lack of clarity. During longitudinal radio-telemetry recording, data were collected over a 10 second window every 10 minutes. Physiological parameters were recorded as follows:

- Body temperature = average Tb recorded over each 10 second sampling window
- heart rate = HR recorded during each 10 second sampling window
- ECG = continuous waveform collected over each 10 second sweep
- Activity = total movement recorded between 10 minute sampling events

Therefore, when the activity recording shows 'no activity', the animal was not active for the entire 10 minute period. For our analyses of 'inactive HR', two consecutive measures of zero activity were required (i.e. 20 minutes of no activity). During acute manipulations (e.g. autonomic blockade), recording frequency was increased to a 10 second sweep every minute. We have now clarified this in the methods section (lines 281-285, 304-305).

14. Provide more details on the software, version, and templates used for the ECG analysis. Also, was analysis done fully automated, manually, or mixed?

We have included additional detail on the ECG analyses in the methods section (lines 282-296). Briefly, radio-telemetry studies were collected and analysed with DSI Dataquest ART4.1 telemetry software. The heart rate was derived directly from the Dataquest acquisition software. Measurements of ECG parameters in longitudinal recordings (i.e. ~5 days of continuous recordings) were conducted with a custom programme written in Matlab (Mathworks). For this analysis, we calculated an ECG waveform template for each individual sweep (the normalised mean of all beats within a pre-defined amplitude window) which was then simply used to identify and remove occasional spurious threshold crossings due to noise (which differed very substantially in shape from the bona fide beats that dominated the average template). We found this approach offered a very robust way of excluding noise while allowing for small changes in the ECG waveform over time and/or between animals. ECG conduction variables were then derived from an automated processing of each identified beat with in-built quality control. Accuracy of the automated analyses were extensively validated by manual assessment of individual sweeps chosen from the data sets. For this, random sweeps were selected from different times of day, different individual animals, and from a variety of QC ratings to ensure correct detection.

15. Line 282: Was injection done i.p., i.v., or otherwise? Was there a vehicle control to control for the effect of the injection itself on stress-induced changes in HR? If not, justify why this was not necessary. What was the time of injection in the different groups?

We have now clarified this in the methods section. Injections were done i.p. (lines 302-304), and control (saline) injections were undertaken one cycle prior to autonomic blockade. All mice showed a similar increase in heart rate in response to saline injection, which returned to pre-injection baseline by 15 minutes post-injection. During autonomic blockade, heart rates were compared 20-30 minutes drug post-administration, and therefore the effects of the injection stress do not interfere

with the interpretation of the results. For increased clarity, we have now included additional text to the methods sections (line 306-308) to highlight this control data.

We have now included additional data demonstrating the effect of complete autonomic block on HR in the 22.5h and 24h LD housed mice at 2h after lights on and 2h after lights off (new manuscript Figure 5k-m). These studies demonstrate tonic repression of heart rate in non-resonant LD housed mice. In contrast, autonomic modulation of HR exhibits a diurnal variation in the 24h LD housed mice (although this did not reach statistical significance).

16. What were the light intensities in the study during L and D, especially in the light exposure studies?

Light intensity recorded during all resonance LD and light pulse experiments was 463 lu^x , and dark (D) was no light. We have now included this information in the methods text (line 255).

17. In panels "a" of Fig. 1, what is the top-to-bottom range in T_b of each line and with what resolution? Is the top-to-bottom range the same for each condition (the five columns) and each day (each line top to bottom)?

The plots shown are derived from individual animal recordings, and are reflective of their respective LD groups. Within the temperature 'actograms', the temperature range differs slightly between animals but are consistent for the duration (day-to-day) of each profile. This is done to highlight the temporal dynamics of the profile (e.g. period and phase) over the long duration study rather than provide information on absolute body temperature. Within each row (double plotted LD cycle) the resolution is 30 minutes (x-axis) and 0.01°C (Y-axis). Importantly, quantitative profiles of body temperature are shown in the manuscript within Figure 1b (group T_b profile across the LD cycle, derived from 10 days of recording for each animal) and Supplementary Figure 1b (which provides individual and group mean T_b).

18. Fig. 2, it is indicated that there were more than 13 animals in each group. It seems there are 7 data points for 27h epWAT. Even though one or two data points may be 'hidden' behind each other, it is unlikely that there are 14 data points. Please clarify.

We apologise for this error, and have altered Figure legend 2 to reflect $n \geq 7$.

19. Fig. 5: Provide statistics for the statement in the Legend: "Mean heart rate (HR) and diurnal rhythms in HR were significantly reduced ...", and clarify what is meant with "reduced... rhythms", presumably "reduced rhythm amplitudes".

This lack of clarity has now been addressed. The figure legend now reads "Mean heart rate (HR) was significantly reduced in non-resonant LD housed mice, which was accompanied by a significant reduction in HR rhythm amplitude in the 27h LD housed mice." Amplitude differences were derived from sine wave fitting of individual animal rhythms, then compared by one-way ANOVA (Supplementary figure 1f).

20. Change the first "d" in the Legend to "c".

This figure legend has been changed.

21. Why is the profile for RR much more smooth compared to that for HR? Is it based on different data sets?

The data depicted in the plots originated from the same study groups. The HR data shown was acquired directly from the Dataquest ART4.1 telemetry software (Data Sciences International) which provides an internally derived measure of heart rate at each sampling interval. The data shown for RR interval was derived from the ECG waveform analyses. Differences in the appearance of the

graphs were likely to have originated from slight differences in beat detection and inclusion criteria between the two. To increase consistency within this data set, we have now included HR profiles derived from the ECG analyses. Importantly, the analyses of HR and RR provided by the two methods are highly consistent in terms of the impact and magnitude of effect of non-resonant LD cycle environments on heart rate.

22. Panel i: which study is this from, how long were the animals exposed to LL, and explain the study rationale.

The LL exposed animals were run in parallel with one of our non-resonant LD studies. The mice were maintained under a 24h LD cycle and subsequently exposed to constant light for 2 weeks, with mean HR derived from the 2nd week of LL exposure. The rationale for the LL study was to confirm published literature, and to provide corroborating evidence that aberrant light exposure causes a depression in HR. As the decrease in HR under constant light confirms published work, and is not of central importance to our studies using non-resonant LD cycles, we have moved this data from the primary to supplementary figures (now Supplementary Figure 3b).

23. "1. HR recorded during the LP was also significantly lower than that recorded the previous light (L) or dark (D) phase." This conclusion is not clear from the data displayed in Fig. S3f. Please explain why these results seem different. Fig. S3: What is meant with "transition period"...? HR is substantially lower the days after the L pulse. How are Tb and activity affected? What is the proposed mechanism that causes this after-effect of the L pulse?

The non-pulsed (NLP) records that were originally shown in Figure 5g and Supplementary Figure 3b/g depict average diurnal profiles generated from the 3 LD cycles preceding the light pulse event. It was this average daytime HR that was compared with the light pulse induced HR, and found to be significantly different (old Figure 5l). The original Supplementary figure 3f depicts continuous recording across the 7 days of the study. As highlighted by the reviewer, when tested against the HR recorded in the light phase immediately preceding the night of the light pulse, HR was not significantly different during the pulse (old Figure 5h). Therefore, we have now removed fig 5l, and have amended the text accordingly (lines 149-152). However, the major impact of acute light pulsing on HR over the following cycles is a highly robust finding, and this striking observation is not affected by the removal of this LP data.

The specificity of the impact of an acute light pulse on HR is further reinforced by the fact that neither body temperature, nor activity was suppressed in the cycles following the acute light pulse (new results added to Supplementary figure 4). Thus, the reduction in HR was not simply a secondary consequence of reduced locomotor activity or an overt disruption of all physiological rhythmicity. The activity profile of the light pulsed mice (Supplementary figure 4a) show clearly that the 2h light pulse caused a phase shift in the mice. The resulting phase misalignment (delayed activity onset in the 2 cycles following the light pulse) was accompanied by a significant reduction in heart rate. This parallels the observations made in the 22.5h LD housed mice, and adds to the evidence that phase misalignment drives a depression of heart rate.

We have clarified the text referring to 'transition period' measures in the figure legend, and have added new data recorded continuously across the transfer of mice from 24h LD to 22.5h LD (Supplementary figure 3a). These results reveal that a significant decrease in HR is evident rapidly when the mice are placed under a non-resonant LD cycle.

Reviewer #2 (Remarks to the Author):

The concept that disturbance of the physiological 12h light cycle in mice leads to physiological disturbance is not new. Thus, the study is of moderate interest and remains merely descriptive. For example, Karatsoreos et al. found that chronic circadian disruption by housing mice in 20-h light/dark cycles results in ...

The reviewer is correct in pointing out that disturbance of the light-dark cycle has been shown to cause physiological disturbance. However, our demonstration that altered phase of entrainment leads to pronounced alteration in energy metabolism and cardiac function is novel and provides new insight to the effects of extreme chronotype and other desynchronising conditions in humans. All of the examples provided by the reviewer utilise 20hr LD cycles (to which the animals cannot entrain). These are valuable and important studies, but detail a very different paradigm of environmental desynchrony.

Minor comment:

Please check your reference list, since ref. 33 and 53 are listed twice.

This error has been corrected.

Reviewer #3 (Remarks to the Author):

Overall, this is a very elegant study which has been well-designed and executed. The data are both extensive and provide a convincing case that the misalignment between internal circadian time and the external light environment underlies key changes in mammalian physiology which have clear implications for human health and disease.

MAIN COMMENTS

1. How was rhythm amplitude affected by these different conditions? Under T20 and T30, rhythms are clearly attenuated, but were any changes in amplitude, light phase activity etc; also observed?

This has been discussed in part in the response to Reviewer 1. As shown in the new Supplementary figure 1, amplitude of body temperature rhythms were not significantly affected in the 22.5 and 27h LD housed mice. The 20 and 30h LD housed mice show strong phase dispersion between the individual mice within the group, and the 20h LD group exhibited reduced temperature rhythm amplitude. Additionally, the 20h and 30h LD housed mice exhibit evidence of light-induced masking and abrupt phase shift. These effects make a robust assessment of light-phase activity difficult.

2. How do the authors reconcile their findings of decreased adiposity with the view that circadian disturbances give rise to increased weight gain and metabolic disease? These findings are not inconsistent – the change in food intake and reliance on carbohydrate substrates may produce changes in behaviour giving rise to different effects if alternative food sources were available (e.g. high fat diet). These differences should be discussed.

Our observations suggest that entrainment to either long or short period LD cycles cause a state of decreased energy efficiency and/or impaired energy storage in the mice. As the reviewer points out, this contrasts with animal studies demonstrating an obesogenic effect of mistimed feeding (Arble et al 2009; Bray et al 2013; Mukherji et al 2016), repeated phase shift (Oike et al 2015; Salgado-Delgado 2010), exposure to dim-light at night (Fonken et al 2010), and a number of clock gene mutation/knockout models. Most studies in which circadian perturbations lead to an increased severity of diet-induced obesity (e.g. the studies listed above) also report a decrease in metabolic rate and/or locomotor activity. We did not observe decreased energy expenditure, body temperature or locomotor activity in the non-resonant animals. This different behavioural response may be partly due to the nutrient source available (as suggested by the reviewer), as well as the nature of the circadian challenge. The mice adapt to the 22.5h and 27h LD cycles in terms of maintaining robust behavioural rhythms, and maintenance of a robust clockwork in the peripheral tissue; yet with an inappropriate phase alignment. Therefore, disruption of the clock function *per se*

may drive an obesogenic bias, whereas a functioning yet mistimed clock leads to energy inefficiency. In an early study, we examined the effect of high fat diet (HFD) on body weight gain in mice housed under 22.5h LD cycles. Over 12 weeks of HFD feeding, no body weight difference was observed between 24h and 22.5h LD housed mice (Response Figure 3). This suggests that any reduction in energy efficiency caused by the phase misalignment was overwhelmed by the increased energy intake. Interestingly, an increase in RER has reported in both mistimed feeding (Bray et al 2013) and dim light at night (Borniger et al 2014) suggesting that this is a common feature misalignment.

Response figure 3. Body weight gain in response to normal chow (NC) or high fat diet (HFD) profile of mice maintained under 24h and 22.5h LD cycle. Final body weight following 13 weeks of NC or HFD feeding in 24h and 22.5h LD housed mice. * = $P < 0.05$, two-way ANOVA, Sidak's post hoc test (n = 8-10/group).

3. In figure 3, 4 and S2, how were differences in phase determined? The CircWave software citation refers to an unpublished methodology, so it is unclear how differences between phase and amplitude under the different conditions were determined.

CircWave is an open source analysis program which uses harmonic regression to fit a sinusoidal wave through the data set, with significance of the rhythm tested against a fit horizontal line. The programme also provides acrophase of the best fit curve, and this was used to generate acrophase comparison plots in Figure 3f. The validity of this analysis method has been demonstrated in a number of published manuscripts (e.g. Oster et al 2006; Keller et al 2009; Malloy et al 2012; Li and Cassone 2015; Leone et al 2015; Santos et al 2015, and many more). Nonetheless, we have verified all of the acrophase measures using JKT Cycle (another commonly used algorithm for identification of rhythmic parameters in gene expression data; Hughes et al 2010 JBR). There were no appreciable differences in the acrophase profiles generated by these two methods (and so we did not alter the original figure). In contrast to the simple acrophase determination, statistical assessment of differences in the rhythm parameters of gene expression profiles under the different LD was carried out by sinusoidal waveform fitting with F test determination of shared characteristics. We have clarified this in the methods and figure legend text.

MINOR COMMENTS

1. In Figure 5j the blue line (no pulse) shows a clear response to the light pulse, whereas the red (pulse) does not. I presume this is mislabelled?

It is not clear what has gone wrong on the reviewer's copy of the figures. The colour coding (red = pulse, and blue = no pulse) is correct on the submitted figures, and reveals a clear decrease in HR during the light pulse. To increase clarity we have added an marker bar and 'LP' on the x-axis to highlight the timing of the pulse. There is a strong suppression of HR during the light pulse and over the following 2 cycles.

2. Line 152-154 refers to Figure S4, which is the CK1 τ data. I presume this should be Fig 3?
This has been amended

REVIEWERS' COMMENTS:

Reviewer #1 (Remarks to the Author):

The authors have sufficiently addressed my raised concerns.

Reviewer #3 (Remarks to the Author):

All comments have now been addressed.